# Concept of Long-Term Sustainable Intralogistics in Plastic Recycling Factory

**Miroslav Fusko \*** , **Monika Bučková, Martin Gašo** , **Martin Krajčovič, Ľuboslav Dulina and Radovan Skokan**

Department of Industrial Engineering, University of Žilina, 010 26 Žilina, Slovak Republic;
monika.buckova@fstroj.uniza.sk (M.B.); martin.gaso@fstroj.uniza.sk (M.G.);
martin.krajcovic@fstroj.uniza.sk (M.K.); luboslav.dulina@fstroj.uniza.sk (Ľ.D.);
radovan.skokan@fstroj.uniza.sk (R.S.)
\* Correspondence: miroslav.fusko@fstroj.uniza.sk; Tel.: +421-41-513-2748

**Abstract:** This paper focuses on the sustainability of material flow and intralogistics in factories, two of the primary keys to productivity. In the past ten years, there has been significant interest in the sustainability of intralogistics to increase productivity. The aim of this article is to investigate the consequences of combining two factories into one, followed by the design of new material flows and intralogistics from a long-term perspective and with respect to sustainability. Therefore, this paper outlines sustainability variants of new material flow solutions in factories. Our methods and approaches were chosen because they represent one of the most rapid ways to rationalise material flows and intralogistics in factories to sustain this concept. In comparison with other projects that eliminate problems in the short term, we have focused on the overall long-term and sustainable concept of intralogistics and material flows over the next 10 to 15 years. Our approach has the potential to increase productivity intralogistics and its long-term sustainability. The proposed variants were verified and evaluated using visTable®touch software (v.2.7, plavis GmbH, Chemnitz, Germany). On the basis of these results, the rational variant of material flows was chosen for the factory. This variant has the highest overall improvement of 25%, even after the implementation of factory two into factory one.

**Keywords:** material flow; intralogistics; sustainability; industrial engineering; rationalised

---

## 1. Introduction

Currently, one trend in factories is finding solutions to problems in intralogistics, material flow, and creating sustainable system areas. A second important trend in factories is sustainability, touching on ecological, economic, and social areas. Its increasing importance for factories can be highlighted because sustainability is considered one of the three top priorities of factory management. The end-to-end optimisation of all materials and information flows lowers costs of internal logistics and increases the security of supply and flexibility. Outdated intralogistics processes often make it necessary to redesign the flow of materials and goods. Historically developed processes block a review of logistic systems and materials in factories. Streamlined material flow is an essential part of a manufacturing factory that strives to compete on the market through the sustainability of its processes. In today's robust competitive environment and worsened economic environment, most factories need to readjust their strategy to produce as many products as possible at a minimal cost. The goal is to make the production process more efficient, encouraging workers to increase sustainability. According to Lorincova et al. [1], factories focus on their employees to gain a competitive edge. Technology, processes, and organisational structure can be copied, but the value that competent and dedicated

employees can bring to factories cannot be easily taken away. As a result, many factories have to cooperate with their employees. Factories must motivate and stimulate employees to make them satisfied with the factory and to prevent them from leaving. Motivated employees help businesses to succeed, as they are more productive. Hence, motivated employees can contribute to making an organization more valuable and profitable.

Using production management methods, a factory can efficiently manage the material flow from the entry of the material into the factory to the final distribution to the customer. Currently, high demands are placed primarily on sustainable production management. Therefore, the factory must adopt a sustainable production management strategy so that it can adapt quickly (business process reconfiguration). One of the main criteria is the perfect functioning of logistics and material flows in the factory because sustainable and continuous production process depends on them [2,3]. Two important reasons for the redesign of facility layouts are the continuous fluctuation of customers' demands and the changing market environment. Changes in the product portfolio, production volume, as well as changes in the manufacturing process and technology can result in lousy utilization of space, overall unsustainability of the system, massive work in progress at a factory, high material handling distances, bottlenecks at workstations, and idle time of facilities and workers [4].

According to Kovács et al. [5], the main objective of a facility layout redesign is to design effective workflow, to improve the productivity of machines, material flow, and workers, as well as the design of such systems' sustainability. Most contemporary factories are facing many dilemmas, many of which are associated with the determination of the ways of acquisition or sustainability of their position in a competitive environment [6]. Factories have to continually increase their economy due to the ever-tougher competition from both domestic and international factories. This means the transition from the market of vendors to the market of purchasers. Both the goals of the market and goals of the factory affect the economy [7].

Today, most factories face many challenges in determining how to gain and maintain their position in a competitive environment at home and abroad. The main challenges are to ensure that manufacturing, supply, and customer processes are well established. These processes transform the input material from one form to another and add value [8,9]. The goal of a factory is to add value efficiently with the least amount of waste in terms of time, material, money, space, and labour as well as shortening the length of material flows. To increase factory productivity and sustainability, these processes and operations must be appropriately selected and arranged to allow a smooth and controlled flow of material through the factory [10,11]. The more efficiently the materials can be produced and converted to the desired products, which operate at the required quality, the more improved the productivity of the factories will be; as a result, the living standard of the employees also will improve. Several scientific papers [12–18] focus on layout design and optimisation of material flow. Each of these publications describe case studies that deal with the optimisation of production systems based on the interpretation of material flow in small and medium-sized factories. The only difference in these case studies is that each publication dealt with the optimisation of material flow in its factory differently and addressed it locally; each publication also used different methods or software solutions. To implement our solution, it was necessary to build on this information and select a software solution that we could use to optimise material flows. We decided to use a software solution from AutoCAD (2018, Autodesk, San Rafael, California), in which we rendered the entire production layout in a 2D environment and then converted it to a visTABLE®touch solution where we could work with the data we collected. There are enough software solutions on the market to optimise material flow. Therefore, the advantage is that if a similar software solution is used, the outputs can be compared.

In our manuscript, we focus on the areas of material flows and intralogistics and the sustainability of these systems. Therefore, in the first chapter, it is necessary to theoretically describe the issue and prepare a literature review and study review from these areas. At the end of the first chapter, we designed the process of sustainable logistics systems design in the factory. This design was verified in a real factory.

### 1.1. Material Flows

According to Khoshnevisan et al. [19], the definition of facility layout may be given as the arrangement of machinery and flow of materials from one facility to another, which minimises material handling costs while considering any physical restrictions on such arrangement. According to Sevigne-Itoiz et al. [20], facility layout considers available space, the final product, user safety, and facility and convenience of operations. Facility layout is concerned with the optimum arrangement of departments with known dimensions in a way that minimises materials handling and ensures effective utilisation of employees, equipment, and space.

On the basis of several definitions of logistics, among its main objectives are the design, optimisation, management, performance, control, and sustainability of material flows. This fact defines the scope of intralogistics activities into three key areas:

- material flow activities in the factory,
- information flow activities in the factory,
- cash flow activities in the factory.

As a result, the fundamental goal of production logistics can be formulated as an effort to maximise the transport capability, reliability, and sustainability of the system with the lowest logistical and production costs [21]. The tasks in the field of material flow optimisation concerning intralogistics have a different feature. Logistic and intralogistics processes have not changed in industrial practice for a long time. Most factories focus their attention on optimising production processes. Production premises are often organised without considering the intralogistics costs of the existing production system. These costs are eliminated or compressed. This results in the following errors: unorganised business processes in factories, long-term unsustainability of the proposed systems, lack of clear assign of responsibilities in individual areas, long transport ways, overlapping ways of different material flows, lack of planned transport ways, unnecessary repackaging processes, vast material reserves in the manufacturing industries, lack of transparency, employee misuse of work time, lack of transparency of intralogistics units, among others. The fundamental precondition for the rationalisation of material flow in production as well as circulation and the design of sustainable operational intralogistics systems is the knowledge of the current state of material flow organisation and management in the factory. The type, quantity, volume, weight, shape, and dimensions of the material affect how it is handled and determine the requirements for handling [22], transport and storage, or, specifically, the packaging of manipulated material or goods. In the case of the material flow analysing, only the most essential material transfers between the points of receipt and delivery of material or goods are identified. A systematic approach to material flow analysis requires obtaining and analysing the information about [23–25]:

- manipulated product,
- manipulated amounts,
- material flow,
- activities ensuring and influencing the material flow,
- time of the individual operations performed with the material.

### 1.2. Three Pillars of Sustainability

The three-pillar concept of sustainability, commonly represented by three intersecting circles (social, economic, and environmental) with overall sustainability at the centre, has become ubiquitous.

The environmental pillar often receives the most attention. Factories are focusing on reducing their carbon footprints, packaging waste, reducing water usage, and their overall effect on the environment. Factories have found that have a beneficial impact on the planet can also have a positive financial impact. Lessening the amount of material used in packaging usually reduces the overall spending on those materials, such as energy savings [26].

The social pillar ties into another poorly defined concept: social license. A sustainable business should have the support and approval of its employees, stakeholders, and the community in which it operates. The approaches to securing and maintaining this support are various, but it comes down to treating employees fairly and being a good neighbour and community member, both locally and globally [26].

The economic pillar of sustainability is where most businesses feel they are on firm ground. That said, profit cannot trump the other two pillars. Profit at any cost is not at all what the economic pillar is about. Activities that fit under the economic pillar include compliance, proper governance, risk management, higher efficiency and productivity, and capital improvements [26].

### 1.3. Review of Intralogistics and Material Flow Studies

The following study review describes what other studies in the field of intralogistics and material flows dealt with and what their aims were.

The study by Garvin et al. [27] does not focus on a specific issue of current concern to Ann Arbor but instead provides a set of recommendations to improve the efficiency of material flows analysed in this study. Their study also provides analytical tools for identifying important flows and prioritising recommendations. After evaluating this study, we concluded that the study only provides recommendations on how to address efficiency improvements, but the aim may not be the sustainability of the system itself, as it will only solve the local problem.

A study by Scholza et al. [28] focused only on an approach for the systematic identification and implementation of lean and resource efficiency potentials focusing on automated guided vehicles. The result was resource efficiency and reduced energy consumption. For sustainable development of production factories, it is necessary to be aware of the energy efficiency of production processes to have a competitive advantage, including material flow processes. After analysing the study, we concluded that energy efficiency is only one small part of the overall sustainability of intralogistics.

The study Krolczyk et al. [8] focused only on current problems of the factory—predominantly on the spatial arrangement of the working stands as well as determination of the internal transport means and the transport tasks. After analysing the study, we determined that we are focusing on a long-term solution to a problem, in contrast to this study.

The research of Bechtsis et al. [29] demonstrates that Automated Guided Vehicles (AGVs) are a rapidly emerging research field with existing studies focusing more on economic ramifications by addressing network optimisation and distribution problems, and less on developing integrated methodological approaches for promoting environmental and social sustainability. This study aims at motivating the role of AGVs as enablers of sustainability in modern manufacturing systems while focusing more on the environmental sustainability echelon. According to our evaluation of this study, only implementing AGVs will not improve the overall system as it is only one part of the improvement.

In a paper by Klumpp et al. [30], they addressed human–computer interaction, a cornerstone for the success of technical innovation in the logistics and supply chain sector. As a major part of social sustainability, this interaction is changing as artificial intelligence applications (Internet of Things, autonomous transport, Physical Internet) are implemented, leading to larger machine autonomy, and hence the transition from a primary executive to a supervisory role of human operators. According to our analysis of the study, if we implement new technologies into chaos, we will only increase the chaos. Therefore, we decided to proceed with a logical rearrangement of material flows.

The studies in Section 1.3 mentioned above focus on the rationalisation and optimisation of intralogistics only statically and try to solve only the local problem. They do not investigate the causes of problems. In our paper, we focused on the overall, long-term, and sustainable concept of intralogistics and material flows applicable for the next 10 to 15 years. Our approach is better because it takes these aspects into account, and by generating many solutions, we have proposed the variant that fully meets the criteria for the long-term sustainability of intralogistics and material flow in a factory.

### 1.4. Description of Sustainable System Creation

Sustainable intralogistics is an essential organisational component of all factories. Increasing its efficiency and reducing operating costs is necessary for maintaining the competitiveness of sustainable factories. This system is one of the critical subsystems of each factory. The aim is the rationalisation and sustainability of transport in the production process using the latest knowledge of logistics theory. We encounter sustainable intralogistics design when changing production technology; changing external conditions can affect the economic benefits of the monitored operating costs. These are mainly focused on energy and fuel consumption as well as the application of new transport technologies. It is possible to evaluate the changes in the sustainability of material flows based on knowledge of the current possibilities of intralogistics. For the design of sustainable intralogistics, it is necessary to apply current knowledge of the logistics theory to become acquainted with the current state of the factory under consideration; the justification of its selection is based on the decision-making process with impact assessment after the proposed changes. The rationalised composition of a sustainable intralogistics system is maintained until the input parameters have been changed based on individual elements of intralogistics that have been selected in the factory [31,32].

According to Pawlewski et al. [33], a factory is part of a dynamic market. The layout "lives" and changes when production lines are closed or newlines are introduced. Managers, production engineers, planners, logisticians, and lean specialists all work on changes in the layout and intralogistics system to achieve its sustainability, competitiveness, and efficiency.

The design of the Concept of Long-term Sustainable Intralogistics, proposed by authors from this article, requires the realization of four primary phases. This algorithm described in the flowchart seen in Figure 1 always re-starts, with a new need to reassess the sustainability of material flows and intralogistics. It is an experimental confirmation and objective demonstration of the suitability of the proposed activities or methods for its intended use.

The first phase is the preparatory phase of the project when it is necessary to define the level of intralogistics and material flows in the factory.

The second phase is the analytical phase. During this phase, a sustainable system begins to form. The project team will be created, responsibilities will be identified, and the necessary analyses of the internal and external environment will be performed.

The third phase is the creation of sustainable material flows and intralogistics. This step begins with the concept of a sustainable system. This basic variant was subsequently verified and validated to determine whether the design meets the requirements for which this sustainable development project was created. As long as it meets the requirements and is in line with the objectives of the project, it is possible to start generating and designing various variants. If it does not meet the requirements, it is necessary to re-create, review, and re-evaluate the analyses and modify the conceptual design accordingly. For the rational generation of design variants, we used the sequence of the next scheme and specific methods of solution, as seen in Figure 2. This step begins with the definition of project limits and limitations. These constraints and limitations are essential to the concept of a sustainable system, as they also affect the generation of variants. After generating the variants, the results are analysed and evaluated using the visTABLE®touch software. On the basis of the statistical results of this software solution plus the technical and economic evaluation of the project, it is possible to select a rational variant that can be implemented into operation. Therefore, it is necessary to choose a validation system to make this process applicable and verifiable.

The fourth phase is the implementation of the designing variants. Furthermore, the most challenging part of this phase is the sustainability of the implemented solution and continuous improvement.

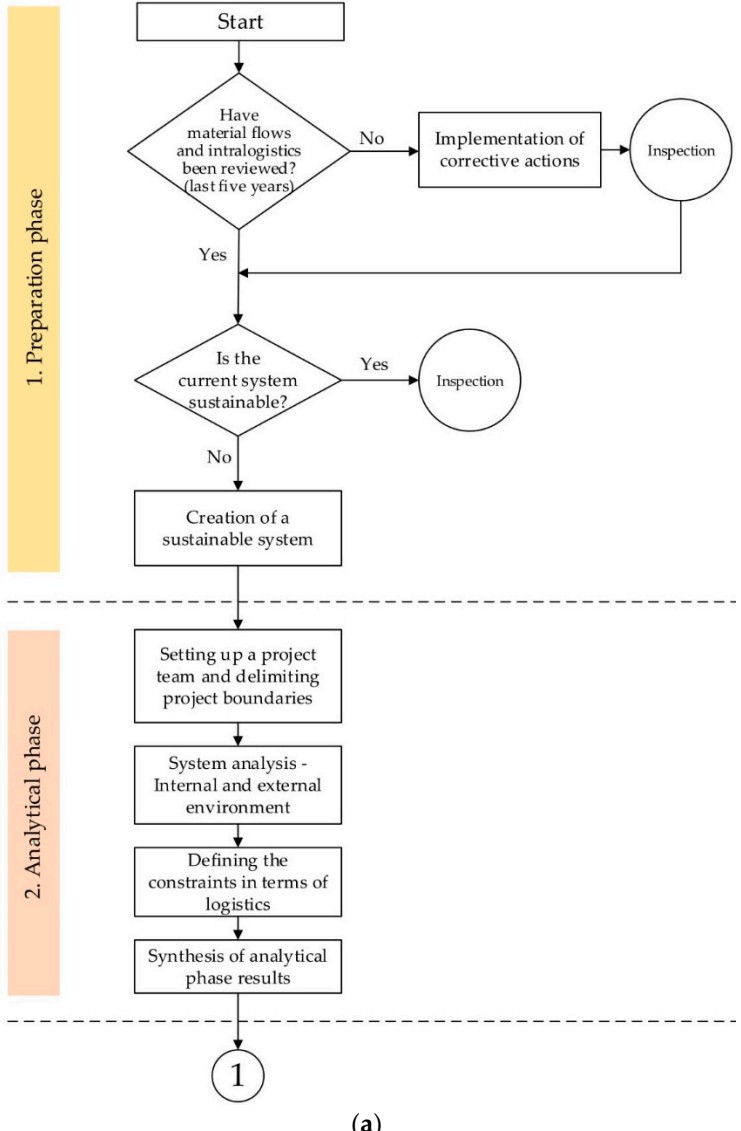

(**a**)

**Figure 1.** *Cont.*

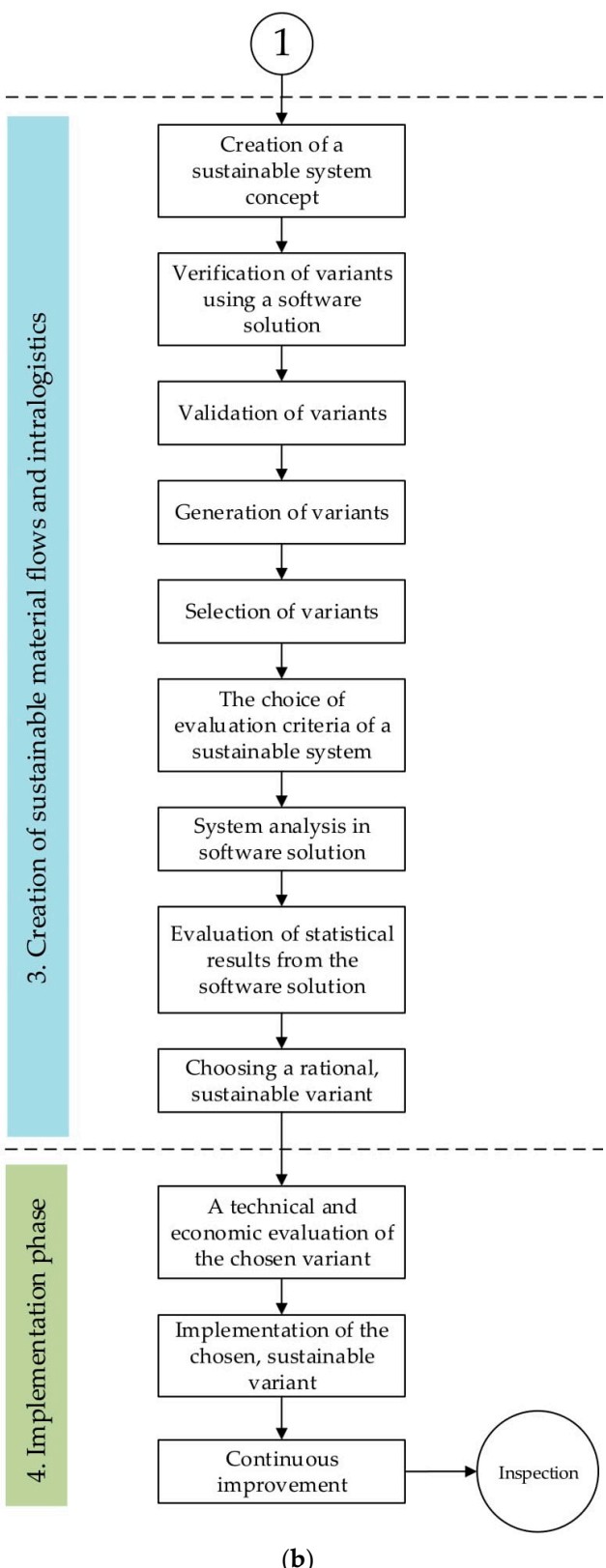

**Figure 1.** The process of design sustainable logistics systems in a factory (**a**,**b**).

The following two figures (Figures 2 and 3) show in detail two critical processes from the third phase (Figure 1)—generation and validation of variants. Specific information from Figures 2 and 3 cannot be published so as to maintain the competitiveness of the factory.

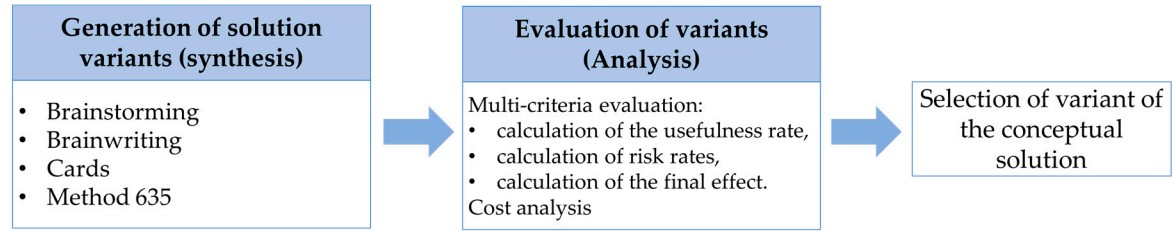

**Figure 2.** Variant design.

Our process, shown in Figure 3, begins with designing and developing systems. At the same time, it is necessary to repeat the process of designing the validation plan and determining responsibilities in the project team to evaluate the current situation of the system continually. Then a process qualification step follows. On the basis of selected key indicators, it is possible to assess and identify critical points of processes that need to be checked. The third step is the implementation of the selected variant, where it is possible to verify selected processes, suggest steps for improvement, choose a new strategy for analysis and evaluation of results.

| Step 1<br>Process design | Step 2<br>Process qualification | Step 3<br>Verification of accuracy |
| --- | --- | --- |
| • Understanding processes<br>• Designing systems in software solutions<br>• Creation of validation plan<br>• Validation responsibility | • Identifying Critical Quality Attributes<br>• Approval and control strategy<br>• Evaluation of processes | • Strategy for analysis and evaluation of results<br>• Monitoring and improvement<br>• Process verification |

**STEPS OF VALIDATION**

**Figure 3.** Steps of validation.

The proposed algorithm (Figure 1) and their steps are designed for the practical designing of a sustainable intralogistics system and sustainable material flows in a factory. We tested this design in a plastic recycling factory. Sections 3 and 4 were carried out under the proposed algorithm and do not reveal proprietary information of the factory.

The share of intralogistics costs account for approximately one-third of total production costs. The choice of a rationally chosen and sustainable transport system and the management of its continuous operation is the responsibility of the intralogistics department in the factory. A simple proof of the intralogistics development in the factory is the ability to offer a competitive pricing strategy for products on the market. Apart from a well-organised in-house and out-of-business transport, competitive pricing strategy is a powerful incentive to find hidden reserves of currently operating logistics systems. By focusing on the sustainability of intralogistics as one of the essential elements of the factory logistics system, the final market position of the product can be influenced. Intralogistics monitors the number of raw materials and materials entering the factory and the number of finished products leaving it, ensuring the continuous flow of transport while maintaining the level of services and time of transportation [34].

## 2. Materials and Methods

The paper deals with empirical research of long-term conceptual solution of organisational, technical, and sustainable ensuring of material flow in a plastic waste processing factory. Sustainable intralogistics in the factory can be characterised as a process of planning, implementing, and coordinating material flow chains and associated information flows at a reasonable cost from the point of origin to the point of consumption through the shortest way. It is also essential to include the subsequent recycling, disposal, and reuse of products. The goal of logistics processes in the factory must be the satisfaction of customers' requirements while also considering environmental and social impacts. Therefore, we had to answer three critical questions in the initial phase of the conceptual design of sustainable material flows and intralogistics in the factory:

1. Where is the factory? (analysis of the processes)
2. What does the factory want to achieve by this change? (goals)
3. How to achieve it? (solution concepts).

By answering these three questions together with the factory owners, we have been able to identify fundamental factory needs and goals for the sustainable growth of the entire factory. To analyse the current state of the factory thoroughly, the project team needed to define the following main areas of the analysis:

- factory products,
- production processes,
- supporting processes,
- management processes,
- costs,
- people,
- time.

The first product of the factory is plastic LDPE (Low-Density Polyethylene) regranulate. Plastic LDPE regranulate is the last stage of ecological recycling of waste plastic foils. It is made of plastic foils waste, which is acquired mainly from the territory of Slovakia, but also from abroad. The picked-up plastic foils are cleaned, sorted fairly, and converted into a plastic melt by heat treatment. In the factory, the Polyethylene (PE) foil is first manually sorted and then sorted by using state-of-the-art technology to ensure perfect cleanliness of the input material. On the basis of many years of experience, the factory produces various types of LDPE foils according to their formula, making their final product the highest quality. The empirical research and progress of our research project on the long-term sustainability of material flows and intralogistics consisted of the steps shown in Figure 4.

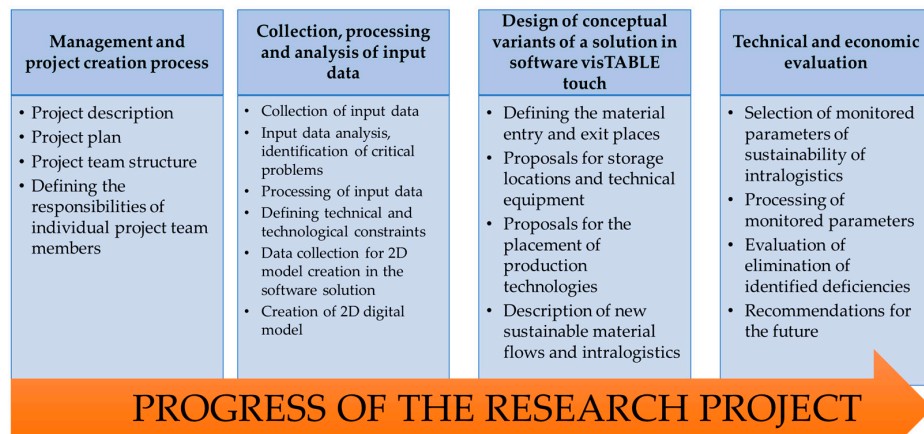

**Figure 4.** Progress of empirical research.

It was necessary to be aware of the fact that we focus on the long-term sustainable aspect of organisational and technical ensuring of material flow. This meant that we had to take into consideration not only the improvement of the current situation but also focus on the trends of industry development that the factory might occur in the next 10 to 15 years.

The studies and the literary sources mentioned above and used in point 1 focus only on the rationalisation and optimisation of intralogistics statically and try to solve only the local problem. They do not investigate the causes of problems. In our paper, we focused on the overall, long-term, and sustainable concept of intralogistics and material flows that might occur in the next 10 to 15 years. The overview of the above studies and literature throughout the first point does not take into account the long-term concept of sustainability of intralogistics and material flows in enterprises. On the other hand, our approach is better because it takes these aspects into account, and by generating many solutions, we have proposed the variant that fully meets the criteria for the long-term sustainability of intralogistics and material flow in the factory. As a result, it will be easier to implement new technologies Industry 4.0 in the future.

In this kind of research project, it is advisable to ask research questions and identify the aim of the work. In our case, the following comprise our research questions:

The first research question aims to determine the impact on the sustainability of the entire production system after the integration of factory two into the factory one.

The second research question aims to propose a long-term sustainable arrangement of material flows and intralogistics in factory one with and without factory two and comparing the improvements or deteriorations. It means that we will compare factory one without or with factory two and their improvements or deteriorations.

The project aims to solve the complex problem of intralogistics and material flows in the long term, i.e., the next 10 to 15 years.

*Software for Realisation of Variants evaluation*

During the research project, we used visTABLE®touch software to create a 2D digital model to analyse data. This software is used for intuitive production planning and logistics systems, optimisation, evaluation, and 2D/3D visualisation of all objects and processes in the manufacturing company. It enables the creation of detailed material flow analysis, optimisation of traffic routes, implementation of value flow analysis, optimisation of logistic processes, and others. VisTABLE®touch software verifies and recalculates any design related to material flow, space requirements, and safety distances in real-time. This feature makes it possible to clarify the depicted layout and practical problems before deciding to incorporate individual designs into reality. The user is provided with visualisation and evaluation functions that are specifically aimed at optimising material flow, such as the Sankey diagram, D–I diagram, aisle utilisation, transport performance and cost, and area analysis. This software creates solutions that optimise material flow and also supports teamwork by making it possible to work on a large-format touch screen, which helps to generate new variants for problem-solving. The D–I diagrams described in Chapter 3 are created using this software.

## 3. Analysis of the Current State in the Factory

Our main goal was to design a long-term concept of organisational, technical, and sustainable ensuring of material flow in a plastic waste processing factory.

Analysis of the intralogistics of the factory under consideration has shown the immense potential of production optimisation with respect to material flow and, consequently, minimisation of the costs involved in it. The problems were the following: the weak concept of intralogistics and the overall long-term unsustainability of the current system concerning the future; frequent production breaks as a result of the incorrect organisation of material delivery and receipt to/from the stands; lack of application of shipping units in transport and storage; significant manual transport work assigned to women employed in the packaging department; "empty" work of internal means of transport (50% of

load during one cycle) for a period more extended than 20% of the working time; operation of the internal transport means with small quantities of materials by forklift trucks; the existence of many reloading points in the production line; too many transport operations; the coincidence of material flow lines; the performance of many unnecessary operations of material handling and transport; as well as differences in delivery time depending on the working personnel (undefined responsibility).

The first partial goal that had to be processed to achieve the main goal was to analyse the current state of ensuring the material flow in the factory. This part of the research project is the first and most important part of the whole project. Without proper knowledge of the current situation, it is not possible to gain the correct results.

As can be seen from Figure 5, the general plan of the factory does not have a suitable layout solution considering the current requirements of manufacturing systems and ancillary and service processes. Individual parts of the factory are located in separate buildings, which are scattered throughout the area, which results in significant and opaque material flows. The ideal material flow should be as short as possible in the form of the letters C, U, or I. These assumptions cannot be reflected in the factory as it exists today.

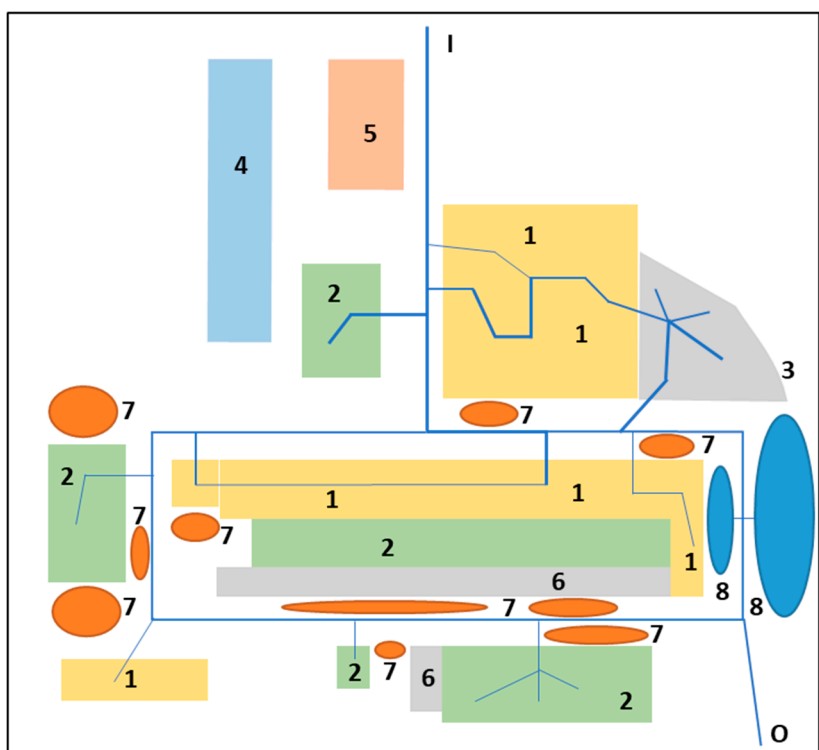

**Figure 5.** Simplified graphical representation of the factory one general plan. I/O—input/output, 1—production, 2—warehouses, 3—expedition, 4—maintenance, 5—administration, 6—other space, 7—stored material—impossible to process immediately, 8—scrap.

Material flows within the production in the factory are displayed by blue lines. The thickness of these lines represents the high intensity of material transport that can be identified in the central part of the factory.

Figure 6 shows the workplaces and their distances from each other according to a distance–intensity (D–I) diagram. The recommended area of the graph is displayed with an orange colour (triangle). In the ideal D–I diagram, the concentration of points near the lower-left corner should be as high as possible. It can be seen from the graph that this is not the case, and therefore it is appropriate to rationalise the layout. The limits in the D–I diagram are based on the current maximums. The maximum for the material flow length currently available is 150 meters.

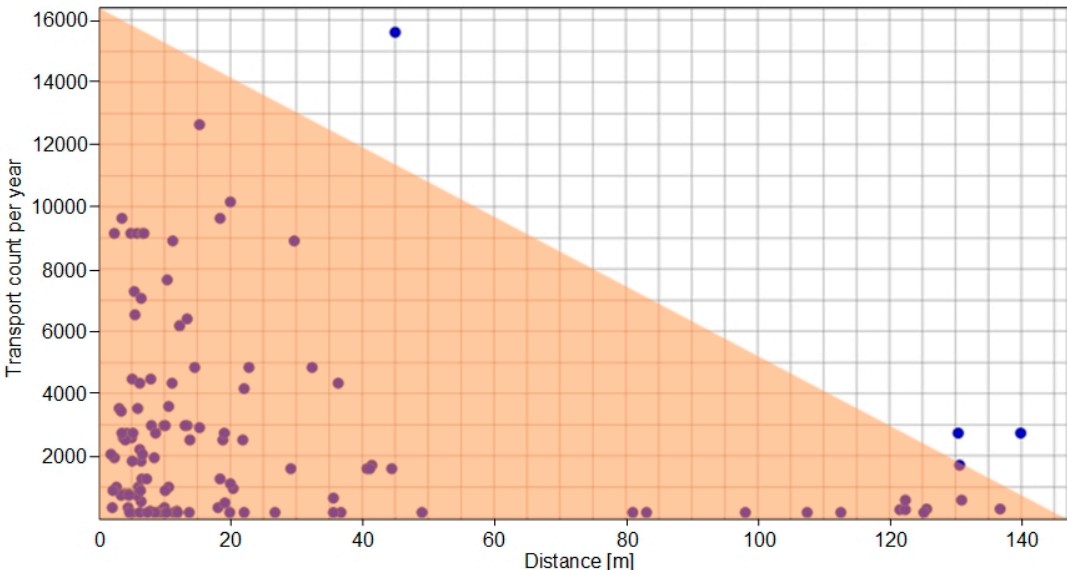

**Figure 6.** Distance–Intensity (D–I) diagram of the current state.

Figure 7 displays a positive approach to inventory reduction, specifically the stock inventory turnover time indicator. Currently, the stock inventory turnover time is approximately 25 days, i.e., the speed/time at which the factory sells on average its stock inventory more precisely goods are in stock on average for 25 days. In comparison with 60 days, this is significant progress to improvement.

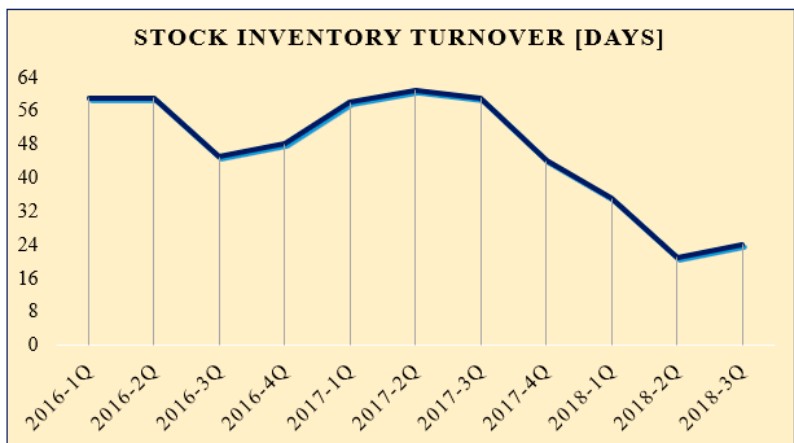

**Figure 7.** Stock inventory turnover time in days. Note: Values are recalculated by a coefficient to keep the exact values confidential.

It can be seen from Figure 8 that, since 2016, the average stock inventories of material have been decreasing. It is a positive signal that the factory is trying to reduce the value of the stock inventories in the long term, which proofs the improvement in the processes in the factory.

Figure 8 is followed by Figure 9, which shows a decreasing trend in the average bound of the financial means in the stock inventories. If the factory has significant financial means in the stock inventories, there is a threat of depreciation.

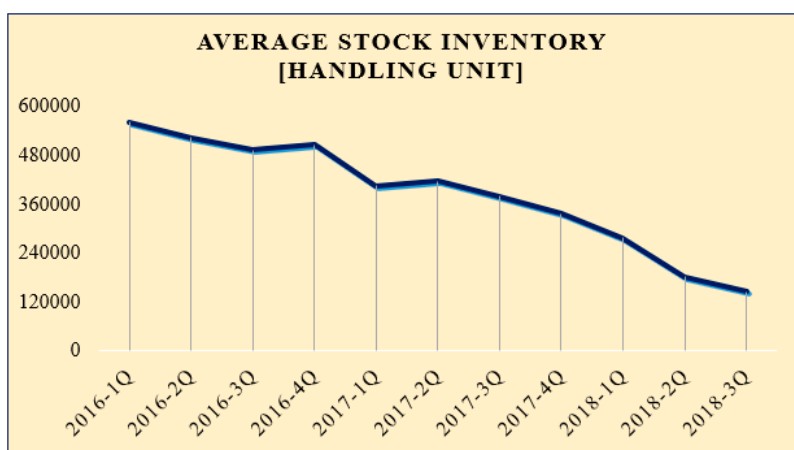

**Figure 8.** Average stock inventory indicator. Note: Values are recalculated by a coefficient to keep the exact values confidential.

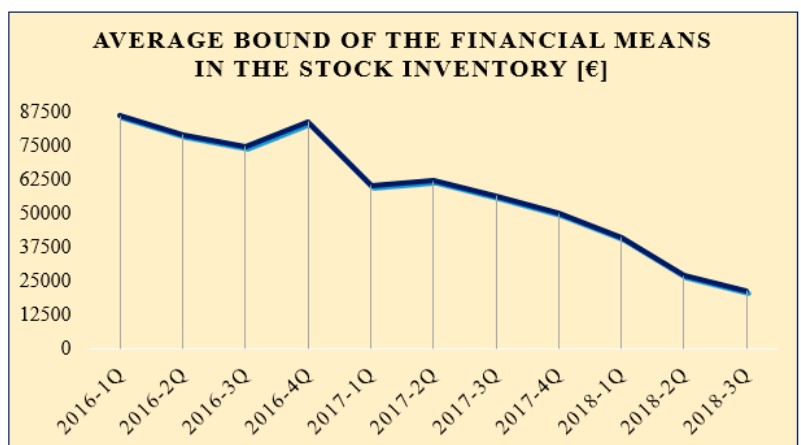

**Figure 9.** Average bound indicator. Note: Values are recalculated by a coefficient to keep the exact values confidential.

The average weekly material receiving is based on an analysis of the past year's history of 57,140.5 handling units, with a standard deviation of 31,150.48 handling units. Taking receiving conditions into consideration, the material mentioned above can be considered as relatively stable. For this reason, we recommend a stock inventory of input material approximately for two weeks of production, which should be enough to ensure the current level of income stability. However, the main recommendation for material receiving is to increase the storage area of the primary material warehouse, which should be a critical factor in determining the stock inventory level. Elimination of other storage areas will have a direct impact on the overall logistics performance and associated costs.

On the basis of the results of data analyses, case processes were mapped during the second phase of the research project, and the main critical issues and development needs of the factory were defined. On the basis of the performed analyses of production processes and related material flows, it was found that the layout of the factory was unsuitable, resulting in complex and lengthy material flows (the average length of material flow from the entry of material into the factory to its output is 33.28 km; total transport capacity per year is 76,171.75 km). Overall, values in these analyses have been recalculated by a coefficient to keep the exact values confidential.

## 4. Results

The current trend in the development of logistics and intralogistics says that only the fastest, the cheapest, and the most efficient productivity in terms of logistics wins. The aim is to save costs where they are most visible, look for improvement, follow trends, and get inspired by the best. The most significant losses in the order flow are logistics in most factories, costs of which often account for more than 30% of total product costs. The area of transport, storage, and handling employs up to 25% of workers, occupies 55% of the area, and makes up 87% of the time spent in the factory.

The different options of possible solutions for material flow and intralogistics in the factory are described in the following four sections. The advantages and disadvantages of each proposed variant are also described.

### 4.1. Variant V1

In the first variant V1 (Figures 10 and 11; Table 1) we were only concerned with the roofing of the dispatch point (arrows), roofing of the space between the gross production (1), the final production (2), and the insertion of the ready-made production of granulate from factory two (3) to the current layout—new roofing. We also moved ready-made garments from factory one to factory two, which produces industrial rubble sacks.

It will be necessary to reserve warehouse space for the production area of factory two, which will move to factory one. If such space is not built, there is a high probability that factory one will have insufficient storage capacity. It is essential to take advantage of the height potential of buildings and think of the shelf-stacker.

Other material flows will not change and will remain as they are today. The variant does not allow the addition of new machines in individual technology centres in the future. New machines would have to be placed on other premises, which would significantly complicate intralogistics and prolong material flows.

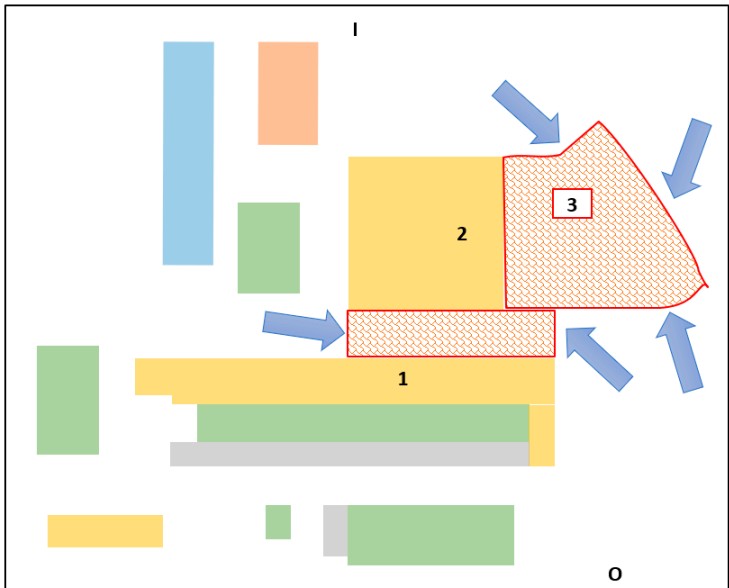

**Figure 10.** Simplified graphical depiction of the variant V1.

One of the main advantages of variant V1 is that the entire production will be concentrated in one site, thus reducing the transport costs between factory one and factory two. The roofing area will be expanded, and the number of areas used to store the input material throughout the factory will be reduced. To use several entry areas for storing input material was not the right solution because the

material flows were long and complicated. Furthermore, the factory has to bear the brunt of the cost of new roofing construction and technology transfer.

**Table 1.** The pros and cons of the variant V1.

| Pros | Cons |
| --- | --- |
| Production in one area. | Long, complex, and opaque material flows. |
| Extension of roofed spaces. | Costs of roofing premises. |
| Transport costs will drop between factory one and factory two. | Transfer costs of technology from the factory two. |
| Reduction of the number on input material areas. | Necessity of reserving storage premises for factory production from factory two. |
| | Sustainable and economical business growth is not ensured. |

D–I diagram of Variant V1

Using the visTABLE®touch software tool, we designed the production process for this type of variant. The software offers us the opportunity to evaluate the solution using the D–I diagram. In Figure 11, we can see that the distance was reduced by 40 m to the resulting value of 110 m in variant V1. We can also see that some workplaces are located outside the zones because their intensity and distance are too high.

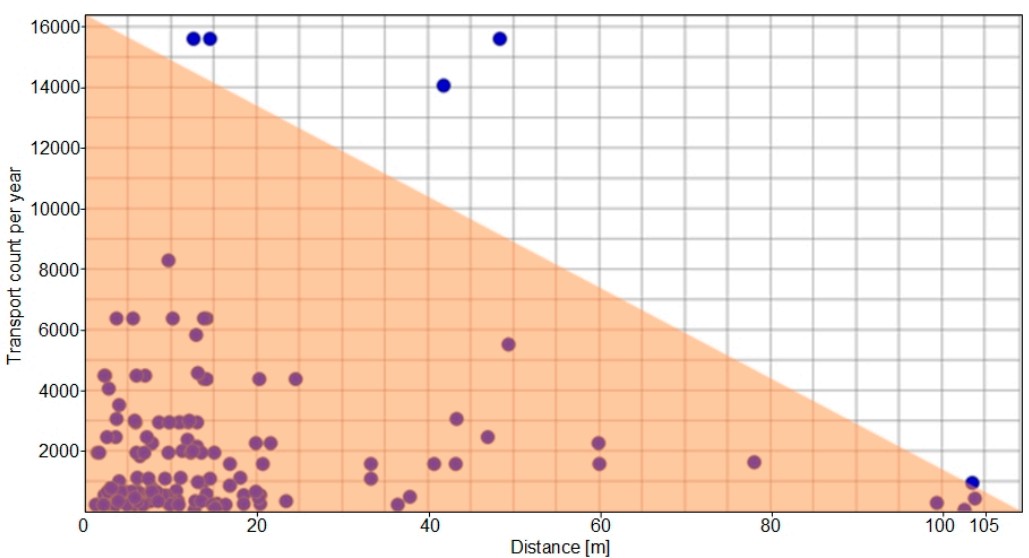

**Figure 11.** D–I diagram of the variant V1.

*4.2. Variant V2*

In the second variant V2 (Figures 12 and 13, Table 2), we can see that the difference between variant V1 and variant V2 is in the rationalisation of the maintenance department. After comparing other advantages of these two variants, we can claim that they remained the same. The necessity to build an entry-exit point oriented to the south with a lifting platform due to the floor spacing can be considered as the main disadvantage of this solution.

In Table 2, we focused on the transfer of the ready-made production of granulate from factory two to the current layout (factory one), to be more specific to the premises of the maintenance department. This activity requires rationalising the maintenance department—sorting and organising the workplace and moving it to other locations. Because the maintenance department is classified as ancillary production processes, it does not have to be located directly next to the production machines and may be placed at the edge of the layout. The new location of the maintenance workshop could be in the rooms at the bottom of the layout.

It will be necessary to reserve warehouse space for the production area of factory two, which will move to factory one. If such space is not built, there is a high probability that there will be insufficient storage capacity. It is essential to take advantage of the height potential of buildings, taking into consideration the shelf-stacker.

Other material flows will not change and will remain as they are today. The variant does not allow the addition of new machines in individual technology centres in the future. New machines would have to be placed on other premises, which would significantly complicate intralogistics and prolong material flows.

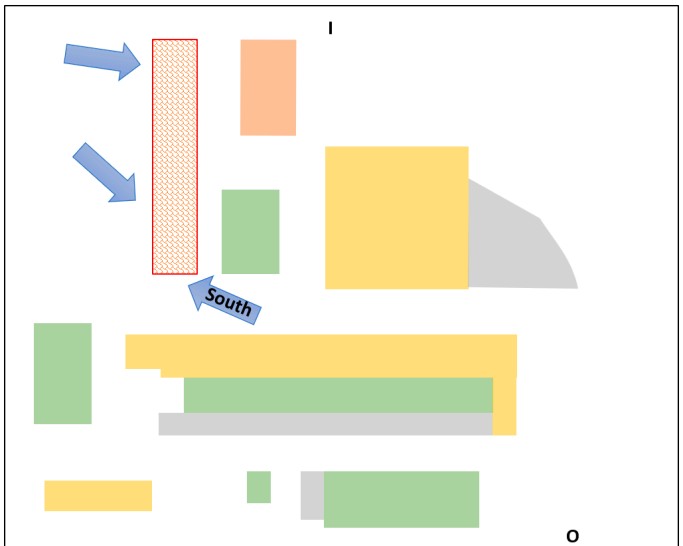

**Figure 12.** Simplified graphical representation of the variant V2.

In Table 2, we can see that the difference between variant V1 and variant V2 is in the rationalisation of the maintenance department. After comparing other advantages of these two variants, we can claim that they remained the same. The necessity to build an entry-exit point oriented to the south with a lifting platform due to the floor spacing can be considered as the main disadvantage of this solution.

**Table 2.** The pros and cons of the variant V2.

| Pros | Cons |
|---|---|
| Production in one area. | Long, complex and opaque material flows. |
| Rationalization of the maintenance department. | Construction of an entry-exit point in the south of the building—a lifting platform. |
| Transport costs will drop between factory one and factory two. | Transfer costs of technology from the factory two. |
| Reduction of the number on input material areas. | Necessity of reserving storage premises for factory production from factory two. |
| | Sustainable and economical business growth is not ensured. |

D–I diagram of the Variant V2

As we can see in variant V2, the D–I diagram has been slightly modified in comparison with variant V1. On the one hand, we still have four production workplaces outside the ideal zone. On the other hand, two of these workplaces no longer have the same intensity as in variant V1. The maximum distance for this variant has been extended by five meters, i.e., 155 meters, and the concentration of workplaces in the left corner slightly increased, which is a positive sign of the reorganisation of workplaces.

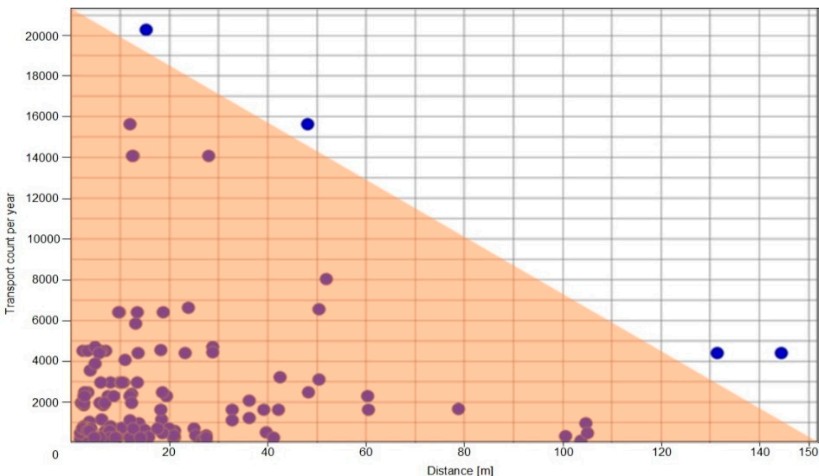

**Figure 13.** D–I diagram of the variant V2.

### 4.3. Variant V3

There are extensive changes in the layout of the third variant V3 (Figures 14 and 15). The foil entering the factory will be unloaded from the truck into the enlarged primary area for the input material (1). This area will be fully roofed and have access to a forklift, which will manipulate with the input material, will move only there. It is also possible to store the input material, which will be processed later. The foil will be transported from the warehouse to gross production by using the forklift (2). All granules will be transported from the granary to an enlarged storage facility (3). The warehouse is designed to take into account the truck turning when passing through the factory.

Furthermore, the regranulation will be transported via forklift through the newly constructed roofing to the space of the former shelf-stacker. The final production (4) will be located here. After the final production, it is necessary to build a store of deflated rolls (5). These pallets will be stored in high shelves due to high utilization of the areas. The deflated foil will be transported by forklift from its warehouse to the ready-made areas—production of final products (6). Finished products will be transported from this area for dispatch in the same way as it is done today (7).

It will be necessary to reserve warehouse space for the production area of factory two, which will move to factory one. If such space is not built, there is a high probability that there will be insufficient storage capacity. It is essential to take advantage of the height potential of buildings and think of the shelf-stacker.

In the future, this variant only allows the addition of new machines for individual technology centres to a limited extent. Some new machines would have to be placed in other areas, which would complicate intralogistics and prolong material flows.

In the variant V3 (Figure 14), we can see that material flows are shortened and straight. As in previous variants, all of production is concentrated in one area. The main advantage (Table 3) of this solution is the removal of the shelf stacker, which is currently not used for storing the regranulate. As a result, the primary area for the input material increased 2.3 times, which allows the truck to pass through the whole factory. Moreover, the input material will be stored only in one place, and it will not be "scattered" over a large part of the factory. In this variant, we were able to enlarge the current large warehouse, where we can store foil for later usage. The primary disadvantages (Table 3) of this variant are the creation of unused space during ready-to-wear and the confection adjustment for the material input warehouse, regranulate storage, transfer of the final production to another hall, construction of a high warehouse for Big Bags and deflated foils, or roofing the area.

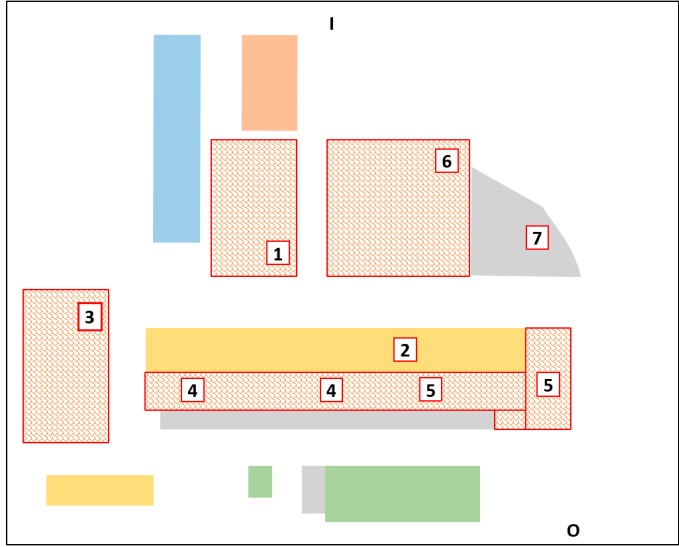

**Figure 14.** Simplified graphical representation of variant V3.

**Table 3.** The pros and cons of variant V3.

| Pros | Cons |
|---|---|
| Shortened and direct material flows. | Creation of unused space in the confection. |
| Complete production in one area. | Costs of space adjustment—material input warehouse, granulate storage, roofing costs. |
| Roofing of the space at the entrance to the shelf-stacker concerning the truck turning. | Costs of transfer of technology from the factory two. |
| Removing the shelf-stacker. | Costs of moving the final production to another hall. |
| Increasing the primary area for the input material by 2.3 times and its roofing. | Costs of building high-rise warehouse for Big Bags. |
| Enlargement of the current large stock of regranulate concerning truck turning. | Costs of building a height warehouse of deflated blown foils. |
| Passage of the truck through the whole factory. | Costs of construction of a high-rise warehouse during roofing before final production. |
| Possibility to store foil, which will be used later in the entrance warehouse area. | The confection is located relatively far from the foil store. |
| Transparency of warehouses and production. | Necessity of reserving storage premises for factory production from factory two. |
| Transport costs will drop between factory one and factory two. | Limited sustainable and economical business growth. |
| Reduction of the number of input material areas. | |

D–I diagram of Variant V3

In this type of variant, we can see that the D–I diagram (Figure 15) has been shortened to 59 m, which is a difference of 91 m from the previous variant. A predominant reason for this is that the shelf stacker has been removed and production has moved there. The disadvantage is that some workplaces are outside of the marked range, in terms of intensity, which is caused by shortening of the material flow up to 91 m.

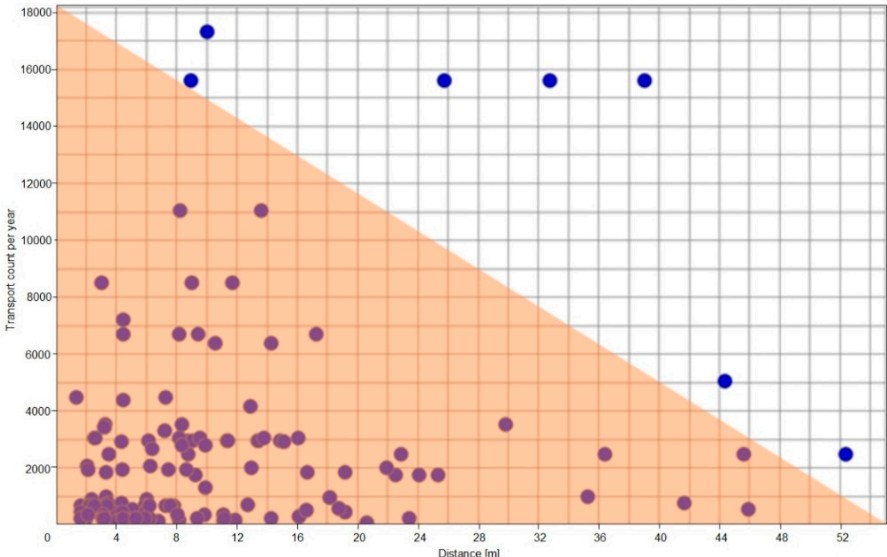

**Figure 15.** D–I diagram of the variant V3.

*4.4. Variant V4*

There are extensive changes in the layout of the fourth variant V4 (Figures 16 and 17; Table 4). The foil entering the factory will be unloaded from the truck into the enlarged primary area used for the input material (1). This area will be fully roofed and forklifted, which will manipulate the input material that will be moved only there. It is also possible to store the input material, which will be processed later. The foil will be transported from the warehouse to gross production by using forklift (2). Total production is rotated by 180 degrees. All granules will be transported from the granary to a high-rise warehouse (3)—roofing between two buildings (entrance shelves). The regranulate (4) inside the hall (5) will be transported (low lift truck, forklift) from the regranulate warehouse (3). The final production is in its original location.

From the final production, the produced foil will travel to the storage area of the deflated foil (6). The warehouse will consist of classic and drive-in racks.

The confection from factory two will be moved to the new space (7). The layout of the machines in the new hall (7) is designed with regard to material flows. The semi-finished products will be transported to the repaired shelf-stacker (8) (forklift, pallet trucks).

The area close to shelf-stacker will be used as the preparation area for shipping (9). Upon the arrival of the truck, the required quantity of products will be shipped. In this variant, technological centres will be created for regranulation, deflating, and ready-made production of granulate rolling, where the individual productions will be separated but will remain close together.

It will be necessary to reserve warehouse space for the production area of factory two, which will move to factory one. If such space is not built, there is a high probability that there will be insufficient storage capacity. It is essential to take advantage of the height potential of buildings and think of the shelf-stacker.

In the future, the variant fully enables the addition of new machines in individual technology centres for the full and sustainable growth of the factory. New machines will be close to each other, intralogistics will be simple, and material flows will be clear. In the future, the shelf-stacker (8) can be removed, and the second regranulator machine can be installed.

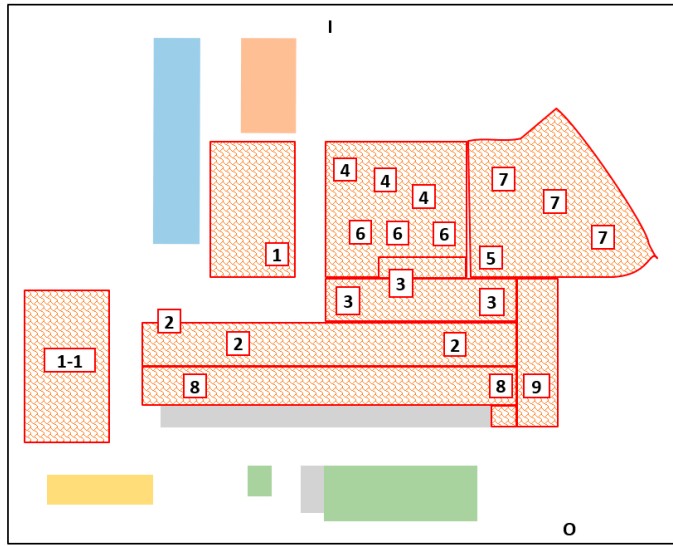

**Figure 16.** Simplified graphical representation of variant V4.

**Table 4.** The pros and cons of variant V4.

| Pros | Cons |
|---|---|
| Shortened and direct material flows. | Creation of unused space in the newly built hall. |
| Complete production in one area. | Costs of adjusting the premises—input material storage. |
| Passage of the truck through the whole area. | Costs of roofing two large spaces. |
| Use of shelf-stacker. | Costs of transfer of technology from factory two. |
| Increasing the primary area for the input material by 2.3 times. | Costs of turning the gross production by 180 degrees. |
| Possibility to store foil used foil later in the area of the entrance warehouse. | Costs of building high-rise warehouse for Big Bags. |
| Construction of technology centres. | Costs of building a height warehouse of deflated foils. |
| Construction of high-rise warehouses. | Shelf-stacker repair. |
| Transparency of warehouses and production. | Necessity of reserving storage premises for factory production from factory two. |
| Transport costs will drop between factory one and factory two. | |
| Reduction of the number of input material areas. | |

The main difference between variant V4 and variant V3 is the use of a shelf-stacker, for which it is necessary to invest eligible money for its repair. Other advantages include the construction of technological centres, construction of high-rise warehouses, transparency of warehouses and production. Minor disadvantages in this variant include unused space, which arises in the newly built hall and rotates the gross production by 180°. Other types of costs remain the same as in the previous variant V3.

D–I diagram of the Variant V4

The difference between Variant V3 and Variant V4 is that we were able to move some production areas closer to the band that is acceptable in the D–I diagram. The total distance in this variant has been reduced by exactly 90 m.

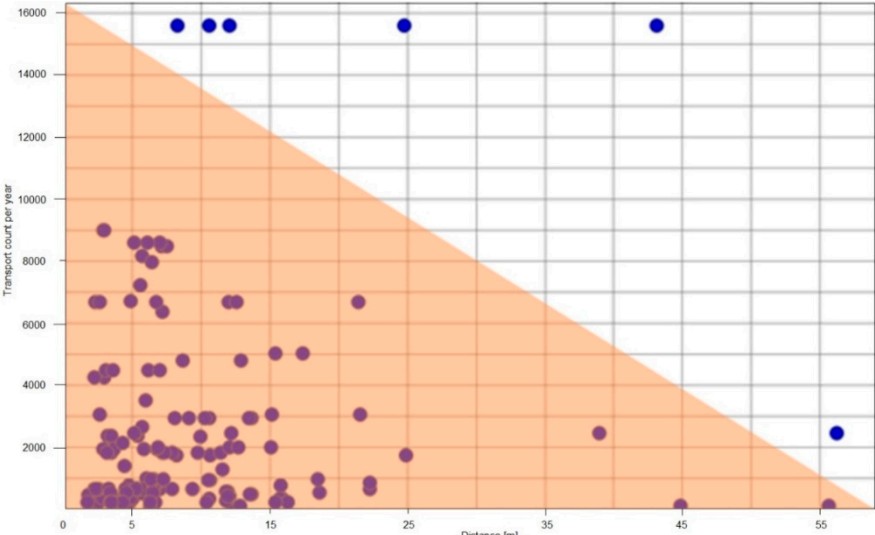

**Figure 17.** D–I diagram of variant V4.

## 5. Discussion

Following the factory strategy, it is necessary to focus on customer requirements and the related need for product delivery at the right time, quality, price and location, the need for differentiation and ecology for the product, such as eco-labelling, elimination of negative impacts on the environment and society, as well as securing the safety of technological equipment operation [35], workers, and elimination of error. The consequences of failures are sometimes severe and relate to safety or the environment; others represent only economic failures [36–38]. Therefore, the concepts of long-term sustainable intralogistics are becoming irreplaceable as an essential tool for the competitiveness of factories. The proposed V4 variant of a sustainable logistics concept for factory material flows will help reduce factory costs, increase factory competitiveness, and consolidate economic growth for the next 10 to 15 years.

The design options consisted of moving technology from factory two to the factory one and implementing it to the current layout. Because of the short-term solution of the organization of material flows, it was appropriate to move factory two to the bottom of the layout of factory one. Warehouses of input material and regranulate would be cancelled at the former positions and built elsewhere. However, in the long run, business development would be inappropriate. The current disposition of the factory is unsatisfactory, but after implementation of the design solutions of the V4 variants, the overall disposition of the factory is rationalized and will be directed to direct material flows.

### 5.1. The Long-Term Perspective of Variant V1

From the perspective of the long-term solution of the organization of material flows in factory one, we proposed the construction and roofing of other external spaces. These areas will be used as production, storage, and partly transport areas. This roofing will also contribute to the better energy efficiency of buildings.

Sustainable and economical business growth is not ensured with this layout.

### 5.2. The Long-Term Perspective of Variant V2

From the perspective of the long-term solution of the organization of material flows in factory one, we proposed using the premises of the former maintenance department in the factory. The entire maintenance department was rationalized and moved to smaller premises. Accordingly, the production from factory two can fit in here. However, the disadvantage of this layout is that direct business growth cannot take place. Production is scattered in the premises of the factory. It is not the right solution.

Sustainable and economical business growth is not ensured with this layout.

### 5.3. The Long-Term Perspective of Variant V3

From the perspective of the long-term solution of the organization of material flows in factory one, we proposed to build a so-called impure and clean part of plastic waste processing; to increase warehouses and for the clean part, we created a production layout with technologies from the factory two at the top of the layout.

With this layout, the sustainable and economic growth of the factory is limited.

### 5.4. The Long-Term Perspective of Variant V4

From the perspective of the long-term solution of the organization of material flows in factory one, we proposed to build a so-called impure and clean part of plastic waste processing increase warehouses, and we created so-called clean part at the top of the layout with the technology from factory two. This layout ensures sustainable and economical business growth. In case of production expansion, it is possible to remove the shelf-stacker and insert the second regranulator. The warehouse space one will serve as the entrance warehouse for regranulator one and warehouse space 1-1 will serve as the entrance warehouse for regranulator two. Then it will be possible to build a new warehouse and shipping areas in the right part of Figure 16. All in all, this option offers the possibility of sustainable business growth, to be more specific of all technology centres and warehouses. Based on knowledge and all evaluations of individual variants, we recommend that the factory implement variant V4. Figure 16 gives a summary comparison of the improvements for each variant. Among the key benefits of the recommended V4 variant belong:

- shortened and direct material flows,
- complete production in one area,
- building technology centres,
- transparency of warehouses and production,
- enabling of high flexibility of further expansion of production in the factory.

From the resulting Tables 5–9 it is clear that the current cost of logistics for the factory is 352,159.17 EUR per year. Each proposed variant defines what investment costs need to be allocated to logistics in the case of implementation of technology from factory two and without factory two. The best economic results were obtained by variant V4, which can reduce logistics costs by 37% per year without factory two or by 25% per year with factory two.

**Table 5.** Economic evaluation of current state. All numerical values are adjusted except for the percentage improvement. Values are recalculated by a coefficient to keep the exact values confidential.

| Property | | Current State |
|---|---|---|
| Total transport capacity | | 76,171.75 km/year |
| Material flow—input-output | | 33.28 km |
| 1 logistics worker | wage 16,637.44 EUR/month | 199,649.29 EUR/year |
| 1 forklift | Fuel | 83,187.21 EUR/year |
| | Maintenance | 69,322.67 EUR/year |
| Total logistics costs per year | | 352,159.17 EUR |

**Table 6.** Economic evaluation of solution variant V1. All numerical values are adjusted except for the percentage improvement of this variant V1. Values are recalculated by a coefficient to keep the exact values confidential.

| Property | | Variant V1 | |
| --- | --- | --- | --- |
| | | Improvement | |
| | | Without Factory 2 | With Factory 2 |
| Total transport capacity | | 62,273.94 km/year | 70,985,03 km/year |
| Material flow—input-output | | 16.22 km | 22.59 km |
| 1 logistics worker | wage 16,637.44 EUR/month | 163,712.42 EUR/year | 185,673.84 EUR/year |
| 1 forklift | Fuel | 68,213.51 EUR/year | 77,364.09 EUR/year |
| | Maintenance | 56,844.58 EUR/year | 64,470.1 EUR/year |
| Total logistics costs per year | | 288,770.51 EUR | 327,508.02 EUR |
| Overall impact on internal logistics performance | | 18% | 7% |

**Table 7.** Economic evaluation of solution variant V2. All numerical values are adjusted except for the percentage improvement of this variant V2. Values are recalculated by a coefficient to keep the exact values confidential.

| Property | | Variant V2 | |
| --- | --- | --- | --- |
| | | Improvement/Deterioration | |
| | | Without Factory 2 | With Factory 2 |
| Total transport capacity | | 67,040.57 km/year | 101,830.84 km/year |
| Material flow—input-output | | 19.41 km | 26.76 km |
| 1 logistics worker | wage 16,637.44 EUR/month | 175,691.38 EUR/year | 267,530.05 EUR/year |
| 1 forklift | Fuel | 73,204.73 EUR/year | 111,470.85 EUR/year |
| | Maintenance | 61,003.95 EUR/year | 92,892.37 EUR/year |
| Total logistics costs per year | | 309,900.06 EUR | 471,893.28 EUR |
| Overall impact on internal logistics performance | | 12% | −34% |

**Table 8.** Economic evaluation of solution variant V3. All numerical values are adjusted except for the percentage improvement of this variant V3. Values are recalculated by a coefficient to keep the exact values confidential.

| Property | | Variant V3 | |
| --- | --- | --- | --- |
| | | Improvement | |
| | | Without Factory 2 | With Factory 2 |
| Total transport capacity | | 57,797.08 km/year | 68,481.09 km/year |
| Material flow—input-output | | 9.71 km | 15.25 km |
| 1 logistics worker | wage 16,637.44 EUR/month | 151,733.46 EUR/year | 179,684,36 EUR/year |
| 1 forklift | Fuel | 63,222.28 EUR/year | 74,868.49 EUR/year |
| | Maintenance | 52,685.23 EUR/year | 62,390.41 EUR/year |
| Total logistics costs per year | | 267,640.96 EUR | 316,943.25 EUR |
| Overall impact on internal logistics performance | | 24% | 10% |

**Table 9.** Economic evaluation of solution variant V4. All numerical values are adjusted except for the percentage improvement of this variant V4. Values are recalculated by a coefficient to keep the exact values confidential.

| Property | | Variant V4 | |
|---|---|---|---|
| | | Improvement | |
| | | Without Factory 2 | With Factory 2 |
| Total transport capacity | | 47,714.79 km/year | 57,496.23 km/year |
| Material flow—input-output | | 8.32 km | 15.25 km |
| 1 logistics worker | wage 16,637.44 EUR/month | 125,779.05 EUR/year | 149,736.97 EUR/year |
| 1 forklift | Fuel | 52,407.94 EUR/year | 62,390.41 EUR/year |
| | Maintenance | 43,673.28 EUR/year | 51,992 EUR/year |
| Total logistics costs per year | | 221,860.27 EUR | 264,119.37 EUR |
| Overall impact on internal logistics performance | | 37% | 25% |

In terms of the proposed methodology and from the perspective of the three pillars of sustainability, we propose to improve the following areas and implement further corrective actions:

- to increase the area for receiving material during administration and roofing the area,
- to enlarge the warehouse in order to store the produced regranulate completely, to build high-rise warehouses,
- to adapt the internal premises of the factory so that the truck does not have to reverse but to ensure a smooth passage of the truck through the factory,
- to implement the 5S methodology indoors and outdoors,
- to implement visual management in the factory,
- renovate the interior and exterior of buildings (building insulation, old windows and doors replacement),
- create two areas in the factory—rough processing and final material processing, thus achieving a cleaner building environment,
- to demolish old unused buildings,
- to build a parking lot, greenery area, or another area for input material at the entrance to the premises,
- the necessity to reserve warehouse space for relocation from the premises two,
- to allocate areas where forklifts can enter and which do not prevent the spread of dirt and heat leakage from buildings,
- the need to replace old diesel and gas (non-ecological) forklifts that moved inside and outside the areas for a new ecological option—on batteries—electric.

After careful evaluation, we have been able to confirm in our study [39], that rationalisation in the factory usually has a positive impact on the processes in the factory. We proved it in this article. The first evidence was the implementation of factory two into factory one (results are shown in Table 5 in Section 5—Discussion). The other evidence was the design of new material flows for intralogistics in factory two (results are shown in Table 5 in Section 5—discussion).

In our case, all three pillars of sustainability are influenced and improved: the social pillar (better labour conditions into the factory, safety improvements, better health conditions), the environmental pillar (emissions reduction, energy savings), and economic pillar (higher efficiency and productivity, sustainable infrastructure development, capital, improvements). In our case, we recommended implementing the V4 variant due to this variant fully reflects on three pillars of sustainability. In this variant, we reduced selected publishable indicators: total transport capacity, material flow, input-output, the wage of one logistics worker, as well as the fuel and maintenance of the forklift. Long-term indicators

of economic benefits as one of the three pillars of sustainability were presented without their numerical values because of the confidentiality of the factory's internal data. The published short-term information of the technical-economic evaluation was comprehensively published above in this article (results are shown in Table 5 in Section 5—Discussion, D–I diagrams of the variants V1–V4).

We also overcame the problems identified at the beginning of a research project on long-term sustainable material flows and intralogistics. Issues that were addressed included: the weak concept of intralogistics and the overall long-term unsustainability of the current system concerning the future, frequent production breaks as a result of the incorrect organisation of material delivery and receipt to/from the stands, lack of application of shipping units in transport and storage, much manual transport work assigned to women employed in the packaging department operation of the internal transport means with small quantities of materials by forklift trucks, the existence of many reloading points in the production line, too many transport operations, the coincidence of material flow lines, performing many unnecessary operations of material handling and transport, as well as differences in delivery time depending on the working personnel (undefined responsibility).

## 6. Conclusions

In our research, we focused on the design of sustainable material flows and intralogistics. We verified our solution while solving the real problems in the factory. The aim was to design material flows rationally with rearrangement of production from the long-term perspective and sustainability. The factory will gradually implement the elements of Industry 4.0 to transform itself into the Factory of the Future (FoF).

A detailed facility layout redesign was completed within the framework of an R and D project. The described case study shows how the efficiency and reduced manufacturing cost of a real-life manufacturing system can be improved by re-layout design, while smaller floor space is needed for the production. The empirical research deals with the conceptual solution of organisational, technical and sustainable material flow, taking into account long term perspective. The main goal of this research was to propose a rational layout of the factory based on rational material flows. The goal of the empirical research regarding the conceptual solution of organisational, technical, and sustainable material flow taking into account long-term perspective was to propose a rational layout of the factory based on rational material flows. In general, the material flow in a factory must represent the organised movement of material (raw materials, work in progress, finished products, and waste) from its entry to exit from the factory. It consists of both passive (material, raw materials, etc.) and active (storage, handling, and transport operations) elements. Addressing the issue and synchronisation of material flows in production and logistics supports the growth of the factory's competitiveness. Therefore, the rational deployment of elements disposition and relationships significantly affects the cost, flexibility, sustainability, growth, and productivity of the entire production system.

**Author Contributions:** All authors contributed to writing the paper in terms of documenting the literature review, analysing the data, and writing the paper. All authors were involved in the finalisation of the submitted manuscript. All authors read and approved the final manuscript.

**Funding:** This work was supported by the Slovak Research and Development Agency under the contract No. APVV-16-0488.

**Acknowledgments:** The authors would like to express their appreciations to the anonymous reviewers and the editors.

**Conflicts of Interest:** The authors declare no conflict of interest.

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
