# Peer review of "Concept of Long-Term Sustainable Intralogistics in Plastic Recycling Factory"

_sustainability, doi:10.3390/su11236750_

Round 1
Reviewer 1 Report
The article deals with the current topic of sustainability of material flow and intralogistics in the factory. The paper describes and compares the current state and four variants of material flow organization and management in a plastic recycling factory. It should be critically noted that the core of the paper seems to be the description of different scenarios.
The area of analysis is relevant; however, I do not see the paper in this current version as a research paper at all but as a project report. The overall paper is designed as a representation of the main results of the comparison of different solutions of material flows and intralogistics for the specific company, without any influence or analysis of existing literature.
The introduction section which should lead from the research context towards a specific research question fails doing so, lacks a logic structure and make blanket statements, which are often not supported by data or references. Moreover, it does not help to clearly understand previous studies and literature gap. A more systematic and critical analysis of literature review is expected. The authors need to explain how this study is built on previous studies to further advance knowledge in this area. Real research questions are not formulated and also the goal of the paper is not derived by theoretical gaps. The novelty of the study is not well argued and lacks substantiation.
The research methodology is not well articulated. The authors need to better describe the methodical approach to finding solutions and validating the findings and justify their choices.
A more in-depth and critical scenarios analysis is also expected. For example, the analysis of the D-I diagram seems to be limited to the maximum distance. Have all the pros and cons been considered in the economic evaluation of solution variants? What about the three pillars of sustainability?
Conclusions are poor in illustrating the value of the paper especially with a theoretical perspective.
Author Response
Dear reviewer,
thank you very much for your comments. Your comments helped us improve the quality of our manuscript. We believe that we've integrated all your comments into the manuscript and met your expectations.
Best Regards
Miroslav Fusko

Reviewer 2 Report
This is a quality paper that has been worked on thoroughly, particularly regarding its technical details. However, there are a few points that need attention before it can be considered for publication:
Abstract: The abstract is expected to include a brief digest of the research, that is, new methods, results, concepts, and conclusions only. The abstract needs to be more focused and achievements needs mentioned clearly. At the moment abstract is more like an introduction than abstract. Please add some information from the conclusion (quantifications).
The strengths and limitations of the applied approach should be clearly identified for the readers of the paper.
Some of the bullet points on the conclusion are simplistic; Please try to emphasize your novelty, put some quantifications, and comment on the limitations. This is a very common way to write conclusions for a learned academic journal. The conclusions should highlight the novelty and advance in understanding presented in the work.
The work seems to be important and should be published in a quality journal, reaching to many scholars in this area. However, this can happen after the above suggestions.
Author Response

(The authors gave the same response as above.)

Round 2
Reviewer 1 Report
The authors have addressed most of the concerns raised regarding the previous version of the manuscript. However, it needs to be improved further to be published.
Line 28: Eliminate “The example of such a variant is the V4 variant.”
Line 84: The last sentence should be contextualized and argued.
Line 96: I would avoid the reference to the figure as it is explained later.
Figure 1: The new figure 1 has significantly improved understanding the design process. However, it could be better arranged. Moreover, I suppose there is a useless label “No”.
Figure 3: I suggest revising the name of Step 3 to be consistent with the others.
Line 123: The rational generation of design variants seems to be included in the second phase, but according to Figure 1 it belongs to the third one. Please, clarify it.
Section 1.3 helps to understand previous studies and literature gap. However, it could be better presented. For example, a critical summary that emphasizes how the proposed study overcomes these limits could be inserted at the end of the section.
Overall, I recommend restructuring the first chapter and improving the connections and transitions between paragraphs. It is not clear if the algorithm described in the flowchart is based on literature or it is a core part of the proposed research work. In the second case, it would deserve a dedicated chapter to give it more emphasis. The “link” between Figure 1, 2, 3 should be better described.
Figure 4: Eliminate “The” in Step 2 title.
Lines 309-312: Revise.
Section 2.2: In the present form, it seems to be a part of the introduction section. Therefore, I suggest moving it to Section 1. On the other hand, these aspects related to the three pillars of sustainability should be more stressed in Section 5. For example, have the mentioned benefits (e.g. 708-711) been considered in the economic evaluation of solution variants? In which way?
Figure 5: Is there a correspondence between the color and the labels of the blocks? If yes, there is probably an error in the gray block 6. Moreover, it is not clear to which factory the layout refers (one, two, both).
A proofreading should be conducted to improve both language and organization quality of the paper.
Author Response
Dear reviewer,
thank you very much for your comments. Your comments helped us improve the quality of our manuscript again. We believe that we've integrated all your comments into the manuscript and met your expectations.
Best Regards
Miroslav Fusko

Round 3
Reviewer 1 Report
The authors have sufficiently addressed the remaining comments regarding the manuscript. However, the quality of the manuscript would gain from a spell check by a native speaker.